# Competitive Distribution Estimation: Why is Good-Turing Good

**Alon Orlitsky**
UC San Diego
alon@ucsd.edu

**Ananda Theertha Suresh**
UC San Diego
asuresh@ucsd.edu

## Abstract

Estimating distributions over large alphabets is a fundamental machine-learning tenet. Yet no method is known to estimate all distributions well. For example, add-constant estimators are nearly min-max optimal but often perform poorly in practice, and practical estimators such as absolute discounting, Jelinek-Mercer, and Good-Turing are not known to be near optimal for essentially any distribution.

We describe the first universally near-optimal probability estimators. For every discrete distribution, they are provably nearly the best in the following two competitive ways. First they estimate every distribution nearly as well as the best estimator designed with prior knowledge of the distribution up to a permutation. Second, they estimate every distribution nearly as well as the best estimator designed with prior knowledge of the exact distribution, but as all natural estimators, restricted to assign the same probability to all symbols appearing the same number of times.

Specifically, for distributions over $k$ symbols and $n$ samples, we show that for both comparisons, a simple variant of Good-Turing estimator is always within KL divergence of $(3 + o_n(1))/n^{1/3}$ from the best estimator, and that a more involved estimator is within $\tilde{\mathcal{O}}_n(\min(k/n, 1/\sqrt{n}))$. Conversely, we show that any estimator must have a KL divergence at least $\tilde{\Omega}_n(\min(k/n, 1/n^{2/3}))$ over the best estimator for the first comparison, and at least $\tilde{\Omega}_n(\min(k/n, 1/\sqrt{n}))$ for the second.

## 1 Introduction

### 1.1 Background

Many learning applications, ranging from language-processing staples such as speech recognition and machine translation to biological studies in virology and bioinformatics, call for estimating large discrete distributions from their samples. Probability estimation over large alphabets has therefore long been the subject of extensive research, both by practitioners deriving practical estimators [1, 2], and by theorists searching for optimal estimators [3].

Yet even after all this work, provably-optimal estimators remain elusive. The add-constant estimators frequently analyzed by theoreticians are nearly min-max optimal, yet perform poorly for many practical distributions, while common practical estimators, such as absolute discounting [4], Jelinek-Mercer [5], and Good-Turing [6], are not well understood and lack provable performance guarantees.

To understand the terminology and approach a solution we need a few definitions. The performance of an estimator $q$ for an underlying distribution $p$ is typically evaluated in terms of the *Kullback-*

*Leibler (KL) divergence* [7],

$$D(p||q) \stackrel{\text{def}}{=} \sum_x p_x \log \frac{p_x}{q_x},$$

reflecting the expected increase in the ambiguity about the outcome of $p$ when it is approximated by $q$. KL divergence is also the increase in the number of bits over the entropy that $q$ uses to compress the output of $p$, and is also the *log-loss* of estimating $p$ by $q$. It is therefore of interest to construct estimators that approximate a large class of distributions to within small KL divergence. We now describe one of the problem's simplest formulations.

## 1.2   Min-max loss

A distribution *estimator* over a support set $\mathcal{X}$ associates with any observed sample sequence $x^* \in \mathcal{X}^*$ a distribution $q(x^*)$ over $\mathcal{X}$. Given $n$ samples $X^n \stackrel{\text{def}}{=} X_1, X_2, \ldots, X_n$, generated independently according to a distribution $p$ over $\mathcal{X}$, the expected KL loss of $q$ is

$$r_n(q, p) = \mathop{\mathbb{E}}_{X^n \sim p^n} [D(p||q(X^n))].$$

Let $\mathcal{P}$ be a known collection of distributions over a discrete set $\mathcal{X}$. The worst-case loss of an estimator $q$ over all distributions in $\mathcal{P}$ is

$$r_n(q, \mathcal{P}) \stackrel{\text{def}}{=} \max_{p \in \mathcal{P}} r_n(q, p), \tag{1}$$

and the lowest worst-case loss for $\mathcal{P}$, achieved by the best estimator, is the min-max loss

$$r_n(\mathcal{P}) \stackrel{\text{def}}{=} \min_q r_n(q, \mathcal{P}) = \min_q \max_{p \in \mathcal{P}} r_n(q, p). \tag{2}$$

Min-max performance can be viewed as regret relative to an oracle that knows the underlying distribution. Hence from here on we refer to it as *regret*.

The most natural and important collection of distributions, and the one we study here, is the set of all discrete distributions over an alphabet of some size $k$, which without loss of generality we assume to be $[k] = \{1, 2, \ldots k\}$. Hence the set of all distributions is the *simplex* in $k$ dimensions, $\Delta_k \stackrel{\text{def}}{=} \{(p_1, \ldots, p_k) : p_i \geq 0 \text{ and } \sum p_i = 1\}$. Following [8], researchers have studied $r_n(\Delta_k)$ and related quantities, for example see [9]. We outline some of the results derived.

## 1.3   Add-constant estimators

The *add-$\beta$* estimator assigns to a symbol that appeared $t$ times a probability proportional to $t+\beta$. For example, if three coin tosses yield one heads and two tails, the add-$1/2$ estimator assigns probability $1.5/(1.5 + 2.5) = 3/8$ to heads, and $2.5/(1.5 + 2.5) = 5/8$ to tails. [10] showed that as for every $k$, as $n \to \infty$, an estimator related to add-3/4 is near optimal and achieves

$$r_n(\Delta_k) = \frac{k-1}{2n} \cdot (1 + o(1)). \tag{3}$$

The more challenging, and practical, regime is where the sample size $n$ is not overwhelmingly larger than the alphabet size $k$. For example in English text processing, we need to estimate the distribution of words following a context. But the number of times a context appears in a corpus may not be much larger than the vocabulary size. Several results are known for other regimes as well. When the sample size $n$ is linear in the alphabet size $k$, $r_n(\Delta_k)$ can be shown to be a constant, and [3] showed that as $k/n \to \infty$, add-constant estimators achieve the optimal

$$r_n(\Delta_k) = \log \frac{k}{n} \cdot (1 + o(1)), \tag{4}$$

While add-constant estimators are nearly min-max optimal, the distributions attaining the min-max regret are near uniform. In practice, large-alphabet distributions are rarely uniform, and instead, tend to follow a power-law. For these distributions, add-constant estimators under-perform the estimators described in the next subsection.

## 1.4 Practical estimators

For real applications, practitioners tend to use more sophisticated estimators, with better empirical performance. These include the Jelinek-Mercer estimator that cross-validates the sample to find the best fit for the observed data. Or the absolute-discounting estimators that rather than add a positive constant to each count, do the *opposite*, and subtract a positive constant.

Perhaps the most popular and enduring have been the *Good-Turing* estimator [6] and some of its variations. Let $n_x \stackrel{\text{def}}{=} n_x(x^n)$ be the number of times a symbol $x$ appears in $x^n$ and let $\varphi_t \stackrel{\text{def}}{=} \varphi_t(x^n)$ be the number of symbols appearing $t$ times in $x^n$. The basic Good-Turing estimator posits that if $n_x = t$,

$$q_x(x^n) = \frac{\varphi_{t+1}}{\varphi_t} \cdot \frac{t+1}{n},$$

surprisingly relating the probability of an element not just to the number of times it was observed, but also to the number other elements appearing as many, and one more, times. It is easy to see that this basic version of the estimator may not work well, as for example it assigns any element appearing $\geq n/2$ times 0 probability. Hence in practice the estimator is modified, for example, using empirical frequency to elements appearing many times.

The Good-Turing Estimator was published in 1953, and quickly adapted for language-modeling use, but for half a century no proofs of its performance were known. Following [11], several papers, e.g., [12, 13], showed that Good-Turing variants estimate the combined probability of symbols appearing any given number of times with accuracy that does not depend on the alphabet size, and [14] showed that a different variation of Good-Turing similarly estimates the probabilities of each previously-observed symbol, and all unseen symbols combined.

However, these results do not explain why Good-Turing estimators work well for the actual probability estimation problem, that of estimating the probability of each element, not of the combination of elements appearing a certain number of times. To define and derive uniformly-optimal estimators, we take a different, competitive, approach.

## 2 Competitive optimality

### 2.1 Overview

To evaluate an estimator, we compare its performance to the best possible performance of two estimators designed with some prior knowledge of the underlying distribution. The first estimator is designed with knowledge of the underlying distribution up to a permutation of the probabilities, namely knowledge of the probability multiset, e.g., $\{.5, .3, .2\}$, but not of the association between probabilities and symbols. The second estimator is designed with exact knowledge of the distribution, but like all *natural estimators*, forced to assign the same probabilities to symbols appearing the same number of times. For example, upon observing the sample $a, b, c, a, b, d, e$, the estimator must assign the same probability to $a$ and $b$, and the same probability to $c$, $d$, and $e$.

These estimators cannot be implemented in practice as in reality we do not have prior knowledge of the estimated distribution. But the prior information is chosen to allow us to determine the best performance of any estimator designed with that information, which in turn is better than the performance of any *data-driven* estimator designed without prior information. We then show that certain variations of the Good-Turing estimators, designed without any prior knowledge, approach the performance of both prior-knowledge estimators for every underlying distribution.

### 2.2 Competing with near full information

We first define the performance of an *oracle-aided* estimator, designed with some knowledge of the underlying distribution. Suppose that the estimator is designed with the aid of an oracle that knows the value of $f(p)$ for some given function $f$ over the class $\Delta_k$ of distributions.

The function $f$ partitions $\Delta_k$ into subsets, each corresponding to one possible value of $f$. We denote the subsets by $P$, and the partition by $\mathbb{P}$, and as before, denote the individual distributions by $p$. Then the oracle knows the unique partition part $P$ such that $p \in P \in \mathbb{P}$. For example, if $f(p)$ is

the multiset of $p$, then each subset $P$ corresponds to set of distributions with the same probability multiset, and the oracle knows the multiset of probabilities.

For every partition part $P \in \mathbb{P}$, an estimator $q$ incurs the worst-case regret in (1),

$$r_n(q, P) = \max_{p \in P} r_n(q, p).$$

The oracle, knowing the unique partition part $P$, incurs the least worst-case regret (2),

$$r_n(P) = \min_q r_n(q, P).$$

The *competitive regret* of $q$ over the oracle, for all distributions in $P$ is

$$r_n(q, P) - r_n(P),$$

the competitive regret over all partition parts and all distributions in each is

$$r_n^{\mathbb{P}}(q, \Delta_k) \stackrel{\text{def}}{=} \max_{P \in \mathbb{P}} \left( r_n(q, P) - r_n(P) \right),$$

and the best possible competitive regret is

$$r_n^{\mathbb{P}}(\Delta_k) \stackrel{\text{def}}{=} \min_q r_n^{\mathbb{P}}(q, \Delta_k).$$

Consolidating the intermediate definitions,

$$r_n^{\mathbb{P}}(\Delta_k) = \min_q \max_{P \in \mathbb{P}} \left( \max_{p \in P} r_n(q, p) - r_n(P) \right).$$

Namely, an oracle-aided estimator who knows the partition part incurs a worst-case regret $r_n(P)$ over each part $P$, and the competitive regret $r_n^{\mathbb{P}}(\Delta_k)$ of data-driven estimators is the least overall increase in the part-wise regret due to not knowing $P$. In Appendix A.1, we give few examples of such partitions.

A partition $\mathbb{P}'$ *refines* a partition $\mathbb{P}$ if every part in $\mathbb{P}$ is partitioned by some parts in $\mathbb{P}'$. For example $\{\{a, b\}, \{c\}, \{d, e\}\}$ refines $\{\{a, b, c\}, \{d, e\}\}$. In Appendix A.2, we show that if $\mathbb{P}'$ refines $\mathbb{P}$ then for every $q$

$$r_n^{\mathbb{P}'}(q, \Delta_k) \geq r_n^{\mathbb{P}}(q, \Delta_k). \tag{5}$$

Considering the collection $\Delta_k$ of all distributions over $[k]$, it follows that as we start with single-part partition $\{\Delta_k\}$ and keep refining it till the oracle knows $p$, the competitive regret of estimators will increase from 0 to $r_n(q, \Delta_k)$. A natural question is therefore how much information can the oracle have and still keep the competitive regret low? We show that the oracle can know the distribution exactly up to permutation, and still the regret will be very small.

Two distributions $p$ and $p'$ *permutation equivalent* if for some permutation $\sigma$ of $[k]$,

$$p'_{\sigma(i)} = p_i,$$

for all $1 \leq i \leq k$. For example, $(0.5, 0.3, 0.2)$ and $(0.3, 0.5, 0.2)$ are permutation equivalent. Permutation equivalence is clearly an equivalence relation, and hence partitions the collection of distributions over $[k]$ into equivalence classes. Let $\mathbb{P}_\sigma$ be the corresponding partition. We construct estimators $q$ that uniformly bound $r_n^{\mathbb{P}_\sigma}(q, \Delta_k)$, thus the same estimator uniformly bounds $r_n^{\mathbb{P}}(q, \Delta_k)$ for any coarser partition of $\Delta_k$, such as partitions into classes of distributions with the same support size, or entropy. Note that the partition $\mathbb{P}_\sigma$ corresponds to knowing the underlying distribution up to permutation, hence $r_n^{\mathbb{P}_\sigma}(\Delta_k)$ is the additional KL loss compared to an estimator designed with knowledge of the underlying distribution up to permutation.

This notion of competitiveness has appeared in several contexts. In data compression it is called *twice-redundancy* [15, 16, 17, 18], while in statistics it is often called *adaptive* or *local minmax* [19, 20, 21, 22, 23], and recently in property testing it is referred as competitive [24, 25, 26] or *instance-by-instance* [27]. Subsequent to this work, [28] studied competitive estimation in $\ell_1$ distance, however their regret is poly$(1/\log n)$, compared to our $\tilde{\mathcal{O}}(1/\sqrt{n})$.

### 2.3 Competing with natural estimators

Our second comparison is with an estimator designed with exact knowledge of $p$, but forced to be *natural*, namely, to assign the same probability to all symbols appearing the same number of times in the sample. For example, for the observed sample $a, b, c, a, b, d, e$, the same probability must be assigned to $a$ and $b$, and the same probability to $c$, $d$, and $e$. Since data-driven estimators derive all their knowledge of the distribution from the data, we expect them to be natural.

We compare the regret of data-driven estimators to that of *natural oracle-aided* estimators. Let $\mathcal{Q}^{\mathrm{nat}}$ be the set of all natural estimators. For a distribution $p$, the lowest regret of a natural estimator, designed with prior knowledge of $p$ is

$$r_n^{\mathrm{nat}}(p) \stackrel{\mathrm{def}}{=} \min_{q \in \mathcal{Q}^{\mathrm{nat}}} r_n(q, p),$$

and the regret of an estimator $q$ relative to the least-regret natural-estimator is

$$r_n^{\mathrm{nat}}(q, p) = r_n(q, p) - r_n^{\mathrm{nat}}(p).$$

Thus the regret of an estimator $q$ over all distributions in $\Delta_k$ is

$$r_n^{\mathrm{nat}}(q, \Delta_k) = \max_{p \in \Delta_k} r_n^{\mathrm{nat}}(q, p),$$

and the best possible competitive regret is $r_n^{\mathrm{nat}}(\Delta_k) = \min_q r_n^{\mathrm{nat}}(q, \Delta_k)$.

In the next section we state the results, showing in particular that $r_n^{\mathrm{nat}}(\Delta_k)$ is uniformly bounded. In Section 5, we outline the proofs, and in Section 4 we describe experiments comparing the performance of competitive estimators to that of min-max motivated estimators.

## 3 Results

Good-Turing estimators are often used in conjunction with empirical frequency, where Good-Turing estimates low probabilities and empirical frequency estimates large probabilities. We first show that even this simple Good-Turing version, defined in Appendix C and denoted $q'$, is uniformly optimal for all distributions. For simplicity we prove the result when the number of samples is $n' \sim \mathrm{poi}(n)$, a Poisson random variable with mean $n$. Let $r_{\mathrm{poi}(n)}^{\mathbb{P}_\sigma}(q', \Delta_k)$ and $r_{\mathrm{poi}(n)}^{\mathrm{nat}}(q', \Delta_k)$ be the regrets in this sampling process. A similar result holds with exactly $n$ samples, but the proof is more involved as the multiplicities are dependent.

**Theorem 1** (Appendix C). *For any $k$ and $n$,*

$$r_{poi(n)}^{\mathbb{P}_\sigma}(q', \Delta_k) \leq r_{poi(n)}^{nat}(q', \Delta_k) \leq \frac{3 + o_n(1)}{n^{1/3}}.$$

Furthermore, a lower bound in [13] shows that this bound is optimal up to logarithmic factors.

A more complex variant of Good-Turing, denoted $q''$, was proposed in [13]. We show that its regret diminishes uniformly in both the partial-information and natural-estimator formulations.

**Theorem 2** (Section 5). *For any $k$ and $n$,*

$$r_n^{\mathbb{P}_\sigma}(q'', \Delta_k) \leq r_n^{nat}(q'', \Delta_k) \leq \tilde{\mathcal{O}}_n\left(\min\left(\frac{1}{\sqrt{n}}, \frac{k}{n}\right)\right).$$

Where $\tilde{\mathcal{O}}_n$, and below also $\tilde{\Omega}_n$, hide multiplicative logarithmic factors in $n$. Lemma 6 in Section 5 and a lower bound in [13] can be combined to prove a matching lower bound on the competitive regret of any estimator for the second formulation,

$$r_n^{\mathrm{nat}}(\Delta_k) \geq \tilde{\Omega}_n\left(\min\left(\frac{1}{\sqrt{n}}, \frac{k}{n}\right)\right).$$

Hence $q''$ has near-optimal competitive regret relative to natural estimators.

Fano's inequality usually yields lower bounds on KL loss, not regret. By carefully constructing distribution classes, we lower bound the competitive regret relative to the oracle-aided estimators.

**Theorem 3** (Appendix D). *For any $k$ and $n$,*

$$r_n^{\mathbb{P}_\sigma}(\Delta_k) \geq \tilde{\Omega}_n\left(\min\left(\frac{1}{n^{2/3}}, \frac{k}{n}\right)\right).$$

### 3.1 Illustration and implications

Figure 1 demonstrates some of the results. The horizontal axis reflects the set $\Delta_k$ of distributions illustrated on one dimension. The vertical axis indicates the KL loss, or absolute regret, for clarity, shown for $k \gg n$. The blue line is the previously-known min-max upper bound on the regret, which by (4) is very high for this regime, $\log(k/n)$. The red line is the regret of the estimator designed with prior knowledge of the probability multiset. Observe that while for some probability multisets the regret approaches the $\log(k/n)$ min-max upper bound, for other probability multisets it is much lower, and for some, such as uniform over 1 or over k symbols, where the probability multiset determines the distribution it is even 0. For many practically relevant distributions, such as power-law distributions and sparse distributions, the regret is small compared to $\log(k/n)$. The green line is an upper bound on the absolute regret of the data-driven estimator $q''$. By Theorem 2, it is always at most $1/\sqrt{n}$ larger than the red line. It follows that for many distributions, possibly for distributions with more structure, such as those occurring in nature, the regret of $q''$ is significantly smaller than the pessimistic min-max bound implies.

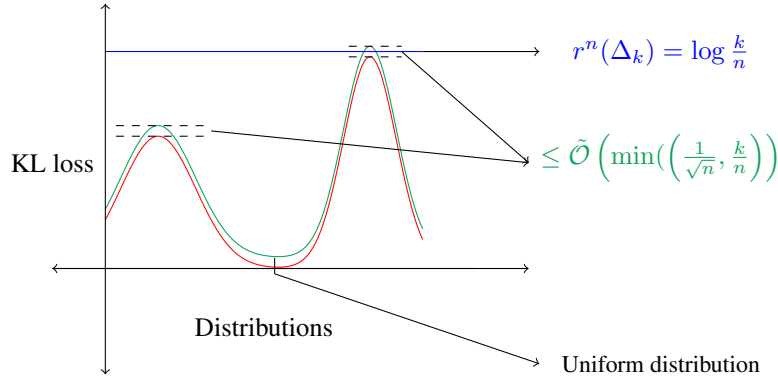

Figure 1: Qualitative behavior of the KL loss as a function of distributions in different formulations

We observe a few consequences of these results.

- Theorems 1 and 2 establish two uniformly-optimal estimators $q'$ and $q''$. Their relative regrets diminish to zero at least as fast as $1/n^{1/3}$, and $1/\sqrt{n}$ respectively, independent of how large the alphabet size $k$ is.

- Although the results are for relative regret, as shown in Figure 1, they lead to estimator with smaller absolute regret, namely, the expected KL divergence.

- The same regret upper bounds hold for all coarser partitions of $\Delta_k$ i.e., where instead of knowing the multiset, the oracle knows some property of multiset such as entropy.

## 4 Experiments

Recall that for a sequence $x^n$, $n_x$ denotes the number of times a symbol $x$ appears and $\varphi_t$ denotes the number of symbols appearing $t$ times. For small values of $n$ and $k$, the estimator proposed in [13] simplifies to a combination of Good-Turing and empirical estimators. By [13, Lemmas 10 and 11], for symbols appearing $t$ times, if $\varphi_{t+1} \geq \tilde{\Omega}(t)$, then the Good-Turing estimate is close to the underlying total probability mass, otherwise the empirical estimate is closer. Hence, for a symbol appearing $t$ times, if $\varphi_{t+1} \geq t$ we use the Good-Turing estimator, otherwise we use the empirical estimator. If $n_x = t$,

$$q_x = \begin{cases} \frac{t}{N} & \text{if } t > \varphi_{t+1}, \\ \frac{\varphi_{t+1}+1}{\varphi_t} \cdot \frac{t+1}{N} & \text{else,} \end{cases}$$

where $N$ is a normalization factor. Note that we have replaced $\varphi_{t+1}$ in the Good-Turing estimator by $\varphi_{t+1} + 1$ to ensure that every symbol is assigned a non-zero probability.

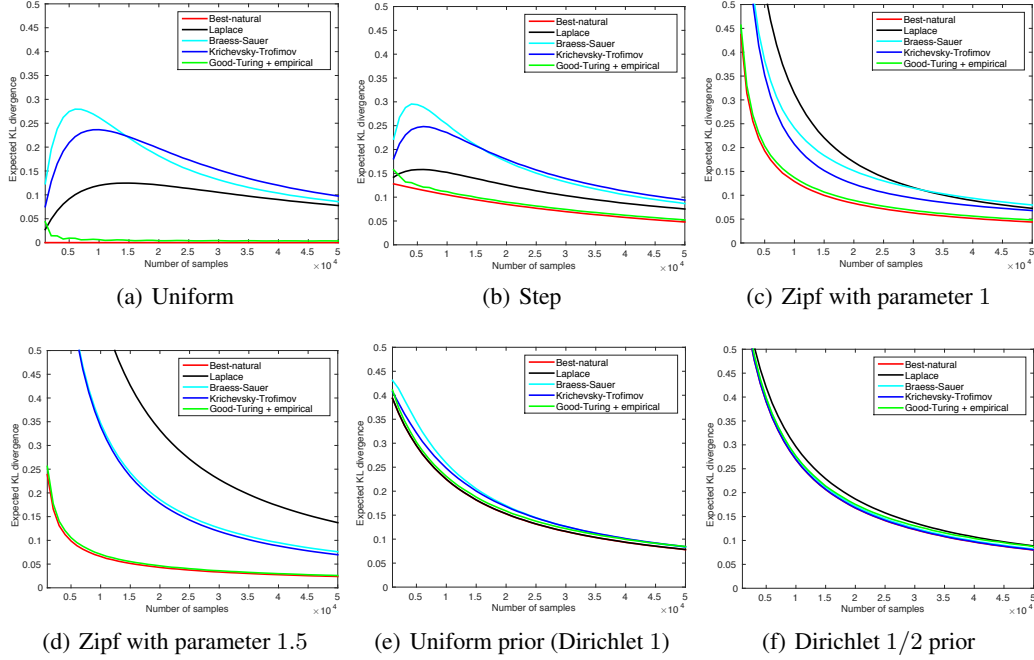

(a) Uniform        (b) Step        (c) Zipf with parameter 1

(d) Zipf with parameter 1.5        (e) Uniform prior (Dirichlet 1)        (f) Dirichlet $1/2$ prior

Figure 2: Simulation results for support 10000, number of samples ranging from 1000 to 50000, averaged over 200 trials.

We compare the performance of this estimator to four estimators: three popular add-$\beta$ estimators and the optimal natural estimator. An add-beta estimator $\hat{S}$ has the form

$$q_x^{\hat{S}} = \frac{n_x + \beta_{n_x}^{\hat{S}}}{N(\hat{S})},$$

where $N(\hat{S})$ is a normalization factor to ensure that the probabilities add up to 1. The Laplace estimator, $\beta_t^L = 1 \forall t$, minimizes the expected loss when the underlying distribution is generated by a uniform prior over $\Delta_k$. The Krichevsky-Trofimov estimator, $\beta_t^{KT} = 1/2 \forall t$, is asymptotically min-max optimal for the cumulative regret, and minimizes the expected loss when the underlying distribution is generated according to a Dirichlet-$1/2$ prior. The Braess-Sauer estimator, $\beta_0^{BS} = 1/2, \beta_1^{BS} = 1, \beta_t^{BS} = 3/4 \forall t > 1$, is asymptotically min-max optimal for $r_n(\Delta_k)$. Finally, as shown in Lemma 10, the optimal estimator $q_x = \frac{S_{n_x}}{\varphi_{n_x}}$ achieves the lowest loss of any natural estimator designed with knowledge of the underlying distribution.

We compare the performance of the proposed estimator to that of the four estimators above. We consider six distributions: uniform distribution, step distribution with half the symbols having probability $1/2k$ and the other half have probability $3/2k$, Zipf distribution with parameter 1 ($p_i \propto i^{-1}$), Zipf distribution with parameter 1.5 ($p_i \propto i^{-1.5}$), a distribution generated by the uniform prior on $\Delta_k$, and a distribution generated from Dirichlet-$1/2$ prior. All distributions have support size $k = 10000$. $n$ ranges from 1000 to 50000 and the results are averaged over 200 trials.

Figure 2 shows the results. Observe that the proposed estimator performs similarly to the best natural estimator for all six distributions. It also significantly outperforms the other estimators for Zipf, uniform, and step distributions.

The performance of other estimators depends on the underlying distribution. For example, since Laplace is the optimal estimator when the underlying distribution is generated from the uniform prior, it performs well in Figure 2(e), however performs poorly on other distributions.

Furthermore, even though for distributions generated by Dirichlet priors, all the estimators have similar looking regrets (Figures 2(e), 2(f)), the proposed estimator performs better than estimators which are not designed specifically for that prior.

# 5 Proof sketch of Theorem 2

The proof consists of two parts. We first show that for every estimator $q$, $r_n^{\mathbb{P}_\sigma}(q, \Delta_k) \leq r_n^{\text{nat}}(q, \Delta_k)$ and then upper bound $r_n^{\text{nat}}(q, \Delta_k)$ using results on combined probability mass.

**Lemma 4** (Appendix B.1). *For every estimator $q$,*

$$r_n^{\mathbb{P}_\sigma}(q, \Delta_k) \leq r_n^{\text{nat}}(q, \Delta_k).$$

The proof of the above lemma relies on showing that the optimal estimator for every class in $P \in \mathbb{P}_\sigma$ is natural.

## 5.1 Relation between $r_n^{\text{nat}}(q, \Delta_k)$ and combined probability estimation

We now relate the regret in estimating distribution to that of estimating the combined or total probability mass, defined as follows. Recall that $\varphi_t$ denotes the number of symbols appearing $t$ times. For a sequence $x^n$, let $S_t \overset{\text{def}}{=} S_t(x^n)$ denote the total probability of symbols appearing $t$ times. For notational convenience, we use $S_t$ to denote both $S_t(x^n)$ and $S_t(X^n)$ and the usage becomes clear in the context. Similar to KL divergence between distributions, we define KL divergence between $S$ and their estimates $\hat{S}$ as

$$D(S||\hat{S}) = \sum_{t=0}^{n} S_t \log \frac{S_t}{\hat{S}_t}.$$

Since the natural estimator assigns same probability to symbols that appear the same number of times, estimating probabilities is same as estimating the total probability of symbols appearing a given number of times. We formalize it in the next lemma.

**Lemma 5** (Appendix B.2). *For a natural estimator $q$ let $\hat{S}_t(x^n) = \sum_{x:n_x=t} q_x(x^n)$, then*

$$r_n^{\text{nat}}(q, p) = \mathbb{E}[D(S||\hat{S})].$$

In Lemma 11(Appendix B.3), we show that there is a natural estimator that achieves $r_n^{\text{nat}}(\Delta_k)$. Taking maximum over all distributions $p$ and minimum over all estimators $q$ results in

**Lemma 6.** *For a natural estimator $q$ let $\hat{S}_t(x^n) = \sum_{x:n_x=t} q_x(x^n)$, then*

$$r_n^{\text{nat}}(q, \Delta_k) = \max_{p \in \Delta_k} \mathbb{E}[D(S||\hat{S})].$$

*Furthermore,*

$$r_n^{\text{nat}}(\Delta_k) = \min_{\hat{S}} \max_{p \in \Delta_k} \mathbb{E}[D(S||\hat{S})].$$

Thus finding the best competitive natural estimator is same as finding the best estimator for the combined probability mass $S$. [13] proposed an algorithm for estimating $S$ such that for all $k$ and for all $p \in \Delta_k$, with probability $\geq 1 - 1/n$,

$$D(S||\hat{S}) = \tilde{\mathcal{O}}_n\left(\frac{1}{\sqrt{n}}\right).$$

The result is stated in Theorem 2 of [13]. One can convert this result to a result on expectation easily using the property that their estimator is bounded below by $1/2n$ and show that

$$\max_{p \in \Delta_k} \mathbb{E}[D(S||\hat{S})] = \tilde{\mathcal{O}}_n\left(\frac{1}{\sqrt{n}}\right).$$

A slight modification of their proofs for Lemma 17 and Theorem 2 in their paper using $\sum_{t=1}^{n} \sqrt{\varphi_t} \leq \sum_{t=1}^{n} \varphi_t \leq k$ shows that their estimator $\hat{S}$ for the combined probability mass $S$ satisfies

$$\max_{p \in \Delta_k} \mathbb{E}[D(S||\hat{S})] = \tilde{\mathcal{O}}_n\left(\min\left(\frac{1}{\sqrt{n}}, \frac{k}{n}\right)\right).$$

The above equation together with Lemmas 4 and 6 results in Theorem 2.

# 6 Acknowledgements

We thank Jayadev Acharya, Moein Falahatgar, Paul Ginsparg, Ashkan Jafarpour, Mesrob Ohannessian, Venkatadheeraj Pichapati, Yihong Wu, and the anonymous reviewers for helpful comments.

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
