[Supplementary Material · supplementary.pdf]

# A    Proofs for competitive formulation

## A.1    Examples of partitions

The following examples evaluate $r_n^{\mathbb{P}}(\Delta_k)$ for the two simplest partitions.

**Example 7.** *The* singleton partition *consists of* $|\Delta_k|$ *parts, each a single distribution in* $\Delta_k$,

$$\mathbb{P}_{|\Delta_k|} \stackrel{def}{=} \{\{p\} : p \in \Delta_k\}.$$

*An oracle-aided estimator that knows the part containing* $p$ *knows* $p$. *The competitive regret of data-driven estimators is therefore the min-max regret,*

$$
\begin{aligned}
r_n^{\mathbb{P}_{|\Delta_k|}}(\Delta_k) &= \min_q \max_{p \in \Delta_k} (r_n(q, \{p\}) - r_n(\{p\})) \\
&= \min_q \max_{p \in \Delta_k} r_n(q, p) \\
&= r_n(\Delta_k),
\end{aligned}
$$

*where the middle equality follows as* $r_n(q, \{p\}) = r_n(q, p)$, *and* $r_n(\{p\}) = 0$.

**Example 8.** *The* whole-collection *partition has only one part, the whole collection* $\Delta_k$,

$$\mathbb{P}_1 \stackrel{def}{=} \{\Delta_k\}.$$

*An estimator aided by an oracle that knows the part containing* $p$ *has no additional information, hence no advantage over a data-driven estimator, and the competitive regret is 0,*

$$
\begin{aligned}
r_n^{\mathbb{P}_1}(\Delta_k) &= \min_q \max_{P \in \{\Delta_k\}} \left( \max_{p \in P} r_n(q, p) - r_n(P) \right) \\
&= \min_q \left( \max_{p \in \Delta_k} r_n(q, p) - r_n(\Delta_k) \right) \\
&= \min_q \max_{p \in \Delta_k} (r_n(q, p)) - r_n(\Delta_k) \\
&= r_n(\Delta_k) - r_n(\Delta_k) \\
&= 0.
\end{aligned}
$$

The examples show that for the coarsest partition of $\Delta_k$, into a single part, the competitive regret is the lowest possible, 0, while for the finest partition, into singletons, the competitive regret is the highest possible, $r_n(\Delta_k)$.

## A.2    Proof of Equation (5)

The definition implies that if $P' \subseteq P$ then $r_n(P') \leq r_n(P)$, for every distribution class $P$ and $P'$. Hence for every $q$,

$$
\begin{aligned}
r_n^{\mathbb{P}'}(q, \Delta_k) &= \max_{P' \in \mathbb{P}'} (r_n(q, P') - r_n(P')) \\
&= \max_{P \in \mathbb{P}} \max_{P \supseteq P' \in \mathbb{P}'} (r_n(q, P') - r_n(P')) \\
&\geq \max_{P \in \mathbb{P}} \max_{P \supseteq P' \in \mathbb{P}'} (r_n(q, P') - r_n(P)) \\
&= \max_{P \in \mathbb{P}} \left( \max_{P \supseteq P' \in \mathbb{P}'} r_n(q, P') - r_n(P) \right) \\
&= \max_{P \in \mathbb{P}} (r_n(q, P) - r_n(P)) \\
&= r_n^{\mathbb{P}}(q, \Delta_k).
\end{aligned}
$$

# B    Upper bounds

For a distribution $p$ and sequence $x^n$, let $p(x^n)$ be the probability of observing $x^n$ under $p$. Recall that for a symbol $x$, we abbreviate $p(x)$ to be $p_x$.

## B.1 Proof of Lemma 4

The proof uses the following result.

**Lemma 9.** *For every class $P \in \mathbb{P}_\sigma$, $r_n(P) \geq \max_{p \in P} r_n^{nat}(p)$.*

*Proof.* We first show that there is an optimal estimator $q$ that is natural. In particular, let

$$q''_y(x^n) = \frac{\sum_{p \in P} p(x^n y)}{\sum_{p' \in P} p'(x^n)}.$$

We show that $q''_y(x^n)$ is an optimal estimator for $P$. Since $q''_y(x^n) = q''_{\sigma(y)}(\sigma(x^n))$ for any permutation $\sigma$, the estimator achieves the same loss for every $p \in P$,

$$\max_{p \in P} r_n(q'', p) = \frac{1}{k!} \sum_{p \in P} r_n(q'', p'). \tag{6}$$

For any estimator $q$,

$$\max_{p \in P} \mathbb{E}[D(p||q)] \overset{(a)}{\geq} \frac{1}{k!} \sum_{p \in P} \mathbb{E}_p[D(p||q)]$$

$$\overset{(b)}{=} \frac{1}{k!} \sum_{p \in P} \sum_{x^n \in \mathcal{X}^n} \sum_{y \in \mathcal{X}} p(x^n y) \log \frac{1}{q_y(x^n)} - H(p)$$

$$= \frac{1}{k!} \sum_{x^n \in \mathcal{X}^n} \sum_{y \in \mathcal{X}} \sum_{p \in P} p(x^n y) \log \frac{1}{q_y(x^n)} - H(p)$$

$$\overset{(c)}{\geq} \frac{1}{k!} \sum_{x^n \in \mathcal{X}^n} \sum_{y \in \mathcal{X}} \sum_{p \in P} p(x^n y) \log \frac{\sum_{p' \in P} p'(x^n)}{\sum_{p'' \in P} p''(x^n y)} - H(p)$$

$$= \frac{1}{k!} \sum_{p \in P} \sum_{x^n \in \mathcal{X}^n} \sum_{y \in \mathcal{X}} p(x^n y) \log \frac{1}{q''_y(x^n)} - H(p)$$

$$\overset{(d)}{=} \frac{1}{k!} \sum_{p \in P} r_n(q'', p).$$

$(a)$ follows from the fact that maximum is larger than the average. $(b)$ follows from the fact that every distribution in $P$ has the same entropy. Non-negativity of KL divergence implies $(c)$. All distributions in $P$ has the same entropy and hence $(d)$. Hence together with Equation (6)

$$r_n(P) = \min_q \max_{p \in P} \mathbb{E}[D(p||q)]$$

$$\geq \frac{1}{k!} \sum_{p \in P} r_n(q'', p')$$

$$= \max_{p \in P} r_n(q'', p).$$

Hence $q''$ is an optimal estimator. Recall that $n_y$ denote the number of times symbol $y$ appears in the sequence. $q''$ is natural as if $n_y = n_{y'}$, then $q''_y(x^n) = q''_{y'}(x^n)$. Since there is a natural estimator that achieves minimum in $r_n(P)$,

$$r_n(P) = \min_q \max_{p \in P} \mathbb{E}[D(p||q)]$$

$$= \min_{q \in \mathcal{Q}^{nat}} \max_{p \in P} \mathbb{E}[D(p||q)]$$

$$\geq \max_{p \in P} \min_{q \in \mathcal{Q}^{nat}} \mathbb{E}[D(p||q)]$$

$$= \max_{p \in P} r_n^{nat}(p),$$

where the last inequality follows from the fact that min-max is bigger than max-min. □

We can now prove Lemma 4.

*Proof of Lemma 4.*

$$r_n^{\mathbb{P}_\sigma}(q, \Delta_k) = \max_{P \in \mathbb{P}_\sigma} \left( \max_{p \in P} \mathbb{E}[D(p||q)] - r_n(P) \right)$$

$$\overset{(a)}{\leq} \max_{P \in \mathbb{P}_\sigma} \left( \max_{p \in P} \mathbb{E}[D(p||q)] - \max_{p \in P} r_n^{\text{nat}}(p) \right)$$

$$\overset{(b)}{\leq} \max_{P \in \mathbb{P}_\sigma} \max_{p \in P} \left( \mathbb{E}[D(p||q)] - r_n^{\text{nat}}(p) \right)$$

$$= \max_{p \in \Delta_k} \left( \mathbb{E}[D(p||q)] - r_n^{\text{nat}}(p) \right)$$

$$= r_n^{\text{nat}}(q, \Delta_k).$$

Lemma 9 implies $(a)$. Difference of maximums is smaller than maximum of differences, hence $(b)$. □

## B.2 Proof of Lemma 5

The proof uses the following lemma which computes the best natural estimator. For a random sequence $X^n$, let $\Phi_t \overset{\text{def}}{=} \varphi_t(X^n)$. Recall that $S_t(x^n)$ is the sum of probabilities of symbols that appears $t$ times in $x^n$. For notational convenience we use $S_t$ to denote both $S_t(x^n)$ and $S_t(X^n)$.

**Lemma 10.** *Let* $q_x^*(x^n) = \frac{S_{n_x}}{\varphi_{n_x}}$, *then*

$$q^* = \arg\min_{q \in \mathcal{Q}^{nat}} r_n(q, p)$$

*and*

$$r_n^{nat}(p) = \mathbb{E}\left[\sum_{t=0}^n S_t \log \frac{\Phi_t}{S_t}\right] - H(p).$$

*Proof.* For a natural estimator $q$, if $n_y = n_{y'}$, then $q_y(x^n) = q_{y'}(x^n)$. Hence, with a slight abuse of notation let $q_{n_y}(x^n) = q_y(x^n)$. For a sequence $x^n$ and estimator $q$,

$$\sum_{y \in \mathcal{X}} p_y \log \frac{1}{q_y(x^n)} - \sum_{t=0}^n S_t \log \frac{\varphi_t}{S_t} = \sum_{t=0}^n \sum_{y: n_y = t} p_y \log \frac{1}{q_y(x^n)} - \sum_{t=0}^n S_t \log \frac{\varphi_t}{S_t}$$

$$= \sum_{t=0}^n S_t \log \frac{1}{q_t(x^n)} - \sum_{t=0}^n S_t \log \frac{\varphi_t}{S_t}$$

$$= \sum_{t=0}^n S_t \log \frac{S_t}{\varphi_t q_t(x^n)}$$

$$\geq 0,$$

where the last inequality follows from the fact that $\sum_{t=0}^n S_t = \sum_{t=0}^n \varphi_t q_t(x^n) = 1$ and KL divergence is non-negative. Furthermore, equality is achieved only by the estimator that assigns $q_x^* = \frac{S_{n_x}}{\varphi_{n_x}}$. Hence,

$$r_n^{\text{nat}}(p) = \min_{q \in \mathcal{Q}^{\text{nat}}} \mathbb{E}\left[\sum_{y \in \mathcal{X}} p_y \log \frac{p_y}{q_y(X^n)}\right] = -H(p) + \mathbb{E}\left[\sum_{t=0}^n S_t \log \frac{\Phi_t}{S_t}\right].$$

□

*Proof of Lemma 5.* As before, with a slight abuse of notation let $q_{n_y}(x^n) = q_y(x^n)$ for natural estimators $q$. For any natural estimator $q$ and sequence $x^n$,

$$\sum_{y \in \mathcal{X}} p_y \log \frac{1}{q_y(x^n)} = \sum_{t=0}^{n} \sum_{y:n_y=t} p_y \log \frac{1}{q_y(x^n)}$$

$$= \sum_{t=0}^{n} S_t \log \frac{S_t}{\varphi_t q_t(x^n)} + \sum_{t=0}^{n} S_t \log \frac{\varphi_t}{S_t}$$

$$= \sum_{t=0}^{n} S_t \log \frac{S_t}{\hat{S}_t} + \sum_{t=0}^{n} S_t \log \frac{\varphi_t}{S_t}.$$

Thus by Lemma 10,

$$r_n^{\text{nat}}(q,p) = -H(p) + \mathbb{E}\left[\sum_{t=0}^{n} S_t \log \frac{S_t}{\hat{S}_t} + \sum_{t=0}^{n} S_t \log \frac{\Phi_t}{S_t}\right] + H(p) - \mathbb{E}\left[\sum_{t=0}^{n} S_t \log \frac{\Phi_t}{S_t}\right]$$

$$= \mathbb{E}\left[\sum_{t=0}^{n} S_t \log \frac{S_t}{\hat{S}_t}\right]$$

$$= \mathbb{E}[D(S||\hat{S})].$$

$\square$

## B.3 Optimality of natural estimators

We now show that exist natural estimators that achieve $r_n^{\text{nat}}(\Delta_k)$ and $r_n^{\mathbb{P}_\sigma}(\Delta_k)$.

**Lemma 11.** *The exists a natural estimator $q''$ such that*

$$r_n^{\text{nat}}(q'', \Delta_k) = r_n^{\text{nat}}(\Delta_k).$$

*Similar there exists a natural estimator $q'$ such that*

$$r_n^{\mathbb{P}_\sigma}(q', \Delta_k) = r_n^{\mathbb{P}_\sigma}(\Delta_k).$$

*Proof.* We prove the result for $r_n^{\text{nat}}(\Delta_k)$. The result for $r_n^{\mathbb{P}_\sigma}(\Delta_k)$ is similar and omitted. Let profile $\bar{\varphi}$ of a sequence $x^n$ be the vector of its prevalences i.e., $\bar{\varphi}(x^n) \overset{\text{def}}{=} (\varphi_0(x^n), \varphi_1(x^n), \varphi_2(x^n), \dots \varphi_n(x^n))$. For any optimal estimator $q$ and sequence $x^n y$ such that $\bar{\varphi}(x^n) = \bar{\varphi}_n$ and $n_y(x^n) = t$, let

$$q''_y(x^n) = \frac{\sum_{w^n z: \bar{\varphi}(w^n) = \bar{\varphi}_n, n_z = t} q_z(w^n)}{\sum_{u^n v: \bar{\varphi}(u^n) = \bar{\varphi}_n, n_v = t} 1}.$$

$q''$ is a natural estimator as if for any sequence $x^n$, $n_y(x^n) = n_{y'}(x^n)$, then $q''_y(x^n) = q''_{y'}(x^n)$. We show that $q''$ is an optimal estimator. Observe that for any $P \in \mathbb{P}_\sigma$

$$r_n(q,P) \overset{(a)}{\geq} \frac{1}{k!} \sum_{p \in P} r_n(q,p) \overset{(b)}{\geq} \frac{1}{k!} \sum_{p \in P} r_n(q'',p) \overset{(c)}{=} r_n(q'',P). \tag{7}$$

Maximum is larger than average and hence $(a)$. Every distribution in $P$ has the same KL loss for $q''$ and hence $(c)$. To prove $(b)$, observe that

$$\sum_{p \in P} r_n(q, p) = \sum_{p \in P} \sum_{x^n \in \mathcal{X}^n} \sum_{y \in \mathcal{X}} p(x^n y) \log \frac{1}{q_y(x^n)} - H(p)$$

$$= \sum_{x^n \in \mathcal{X}^n} \sum_{y \in \mathcal{X}} \sum_{p \in P} p(x^n y) \log \frac{1}{q_y(x^n)} - H(p)$$

$$= \sum_{\bar{\varphi}_n, t} \sum_{x^n : \bar{\varphi}(x^n) = \bar{\varphi}_n} \sum_{y : n_y = t} \sum_{p \in P} p(x^n y) \log \frac{1}{q_y(x^n)} - H(p)$$

$$\overset{(d)}{\geq} \sum_{\bar{\varphi}_n, t} \sum_{x^n : \bar{\varphi}(x^n) = \bar{\varphi}_n} \sum_{y : n_y = t} \sum_{p \in P} p(x^n y) \log \frac{\sum_{u^n, v : \bar{\varphi}(u^n) = \bar{\varphi}_n, n_v = t} 1}{\sum_{w^n, z : \bar{\varphi}(w^n) = \bar{\varphi}_n, n_z = t} q_z(w^n)} - H(p)$$

$$= \sum_{\bar{\varphi}_n, t} \sum_{x^n : \bar{\varphi}(x^n) = \bar{\varphi}_n} \sum_{y : n_y = t} \sum_{p \in P} p(x^n y) \log \frac{1}{q''_y(x^n)} - H(p)$$

$$= \sum_{p \in P} r_n(q'', p),$$

For all sequences $x^n y$ with the same $\bar{\varphi}(x^n)$ and $n_y(x^n)$, $\sum_{p \in P} p(x^n y)$ is the same. Hence, applying log-sum inequality results in $(d)$. By Lemma 10, every $p \in P$ has the same $r_n^{\text{nat}}(p)$, hence subtracting $r_n^{\text{nat}}(p)$ from both sides of Equation (7) results in

$$\max_{p \in P} \left( r_n(q, p) - r_n^{\text{nat}}(p) \right) \geq \max_{p \in P} \left( r_n(q'', p) - r_n^{\text{nat}}(p) \right).$$

Hence for the optimal estimator $q$,

$$r_n^{\text{nat}}(\Delta_k) = \max_{p \in \Delta_k} \left( r_n(q, p) - r_n^{\text{nat}}(p) \right)$$

$$= \max_{P \in \mathbb{P}_\sigma} \left( \max_{p \in P} \left( r_n(q, p) - r_n^{\text{nat}}(p) \right) \right)$$

$$\geq \max_{P \in \mathbb{P}_\sigma} \left( \max_{p \in P} \left( r_n(q'', p) - r_n^{\text{nat}}(p) \right) \right)$$

$$= \max_{p \in \Delta_k} \left( r_n(q'', p) - r_n^{\text{nat}}(p) \right)$$

$$= r_n(q'', \Delta_k).$$

Thus $q''$ is an optimal estimator and furthermore it is natural, hence the lemma. $\qquad \square$

## C   Regret bounds on the Good-Turing estimator

### C.1   Preliminaries

In practice, often the Good-Turing estimator is used for small multiplicities and empirical estimators are used for large multiplicities. We analyze this estimator and bound its regret. For a symbol appearing $t$ times, we assign probability $q'_x = \hat{S}_t / \varphi_t$, where $\hat{S}_t = C_t / N$. $N$ is the normalization factor to ensure that $\sum_{t=0}^{\infty} \hat{S}_t = 1$ and

$$C_t = \begin{cases} \varphi_t \cdot \frac{t}{n} & \text{if } t \geq t_0, \\ (\varphi_{t+1} + 1) \cdot \frac{t+1}{n} & \text{else.} \end{cases}$$

We set $t_0 \propto n^{1/3}$ later. Similar to our experiments, we have modified the Good-Turing estimator to $(\varphi_{t+1} + 1) \cdot \frac{t+1}{n}$, thus ensuring that we never assign a non-zero probability. However, unlike our experiments, where we decided between empirical and Good-Turing estimators depending on if $\varphi_{t+1} \geq t$, for our proofs we just decide it based on $t$ for convenience. We remark that in our experiments the estimator in Section 4 performed better than the one above.

Ideally we would like to analyze this estimator when the number of samples is $n$. However, such analysis is complicated as the number of times symbols appear are dependent, for example, they add to $n$. A standard approach to overcome the dependence, e.g., [29], samples the distribution a random number of times $\sim \text{poi}(n)$, the Poisson distribution with parameter $n$. Some useful properties of Poisson sampling include: $(i)$ A symbol with probability $p$ appears $\text{poi}(np)$ times, $(ii)$ The numbers of times different symbols appear are independent of each other, $(iii)$ For any fixed $n_0$, conditioned on the length $\text{poi}(n) \geq n_0$, the distribution of the first $n_0$ elements is identical to sampling $p$ i.i.d. exactly $n_0$ times. Thus, to simplify the analysis of the estimator, we assume that the number of samples is a Poisson random variable with mean $n$. A similar result holds with $n$ samples.

We first relate the KL regret to a chi-squared like distance between $S$ and $C$.

**Lemma 12.** *For any distribution $p \in \Delta_k$,*

$$\mathbb{E}[D(S||\hat{S})] \leq \sum_{t=0}^{t_0-1} \mathbb{E}\left[\frac{(S_t - (t+1)(\Phi_{t+1}+1)/n)^2}{(\Phi_{t+1}+1)(t+1)/n}\right] + \sum_{t=t_0}^{\infty} \mathbb{E}\left[\frac{(S_t - t\Phi_t/n)^2}{\Phi_t t/n}\right].$$

*Proof.* Since $\log(1+y) \leq y$, $\sum_{t=0}^{\infty} S_t = 1$, and $\sum_{t=0}^{\infty} C_t = N$,

$$D(S||\hat{S}) = \sum_{t=0}^{\infty} S_t \log \frac{S_t}{\hat{S}_t}$$

$$= \sum_{t=0}^{\infty} S_t \log \frac{NS_t}{C_t}$$

$$= \sum_{t=0}^{\infty} S_t \log \frac{S_t}{C_t} + \sum_{t=0}^{\infty} S_t \log N$$

$$= \sum_{t=0}^{\infty} S_t \log \left(1 + \frac{S_t - C_t}{C_t}\right) + \log N$$

$$\leq \sum_{t=0}^{\infty} S_t \left(\frac{S_t - C_t}{C_t}\right) + \log N$$

$$= \sum_{t=0}^{\infty} (S_t - C_t)\left(\frac{S_t - C_t}{C_t}\right) + \sum_{t=0}^{\infty} C_t \left(\frac{S_t - C_t}{C_t}\right) + \log N$$

$$= \sum_{t=0}^{\infty} (S_t - C_t)\left(\frac{S_t - C_t}{C_t}\right) + \sum_{t=0}^{\infty} (S_t - C_t) + \log N$$

$$= \sum_{t=0}^{\infty} \frac{(S_t - C_t)^2}{C_t} + 1 - N + \log N$$

$$\leq \sum_{t=0}^{\infty} \frac{(S_t - C_t)^2}{C_t}$$

$$= \sum_{t=0}^{t_0-1} \frac{(S_t - C_t)^2}{C_t} + \sum_{t=t_0}^{\infty} \frac{(S_t - C_t)^2}{C_t}.$$

Taking expectations on both sides and substituting $C_t$ results in the lemma. □

### C.2 Empirical estimators

All of our results including the next lemma hold for all distributions in $\Delta_k$ and hence stated without any condition on the underlying distribution. Let $N_x \overset{\text{def}}{=} n_x(X^n)$ for a random sequence $X^n$.

**Lemma 13.** *For any $n$ and $t_0$,*

$$\sum_{t=t_0}^{\infty} \mathbb{E}\left[\frac{(S_t - t\Phi_t/n)^2}{t\Phi_t/n}\right] \leq \frac{1}{t_0}.$$

*Proof.*

$$\sum_{t=t_0}^{\infty} \frac{(S_t - t\Phi_t/n)^2}{t\Phi_t/n} \leq \sum_{t=t_0}^{\infty} \frac{(S_t - t\Phi_t/n)^2}{\Phi_t t_0/n}$$

$$\overset{(a)}{\leq} \sum_{t=t_0}^{\infty} \sum_x 1_{N_x=t} \frac{(p_x - t/n)^2}{t_0/n}$$

$$= \sum_x \sum_{t=t_0}^{\infty} 1_{N_x=t} \frac{(p_x - t/n)^2}{t_0/n}$$

$$\leq \sum_x \sum_{t=0}^{\infty} 1_{N_x=t} \frac{(p_x - t/n)^2}{t_0/n}.$$

$(a)$ follows from the fact that $\frac{(\sum_{x=1}^m a_x)^2}{m} \leq \sum_{i=1}^m a_x^2$ for $a_x = 1_{N_x=t}(p_x - t/n)$ and $m = \Phi_t$. Taking expectations on both sides,

$$\sum_{t=t_0}^{\infty} \mathbb{E}\left[\frac{(S_t - t\Phi_t/n)^2]}{\Phi_t t/n}\right] \leq \sum_x \frac{\mathbb{E}[\sum_{t=0}^{\infty} 1_{N_x=t}(p_x - t/n)^2]}{t_0/n}$$

$$\leq \sum_x \frac{p_x/n}{t_0/n}$$

$$= \frac{1}{t_0},$$

where the second inequality follows from observing that $\mathbb{E}[\sum_{t=0}^{\infty} 1_{N_x=t}(p_x - t/n)^2]$ is the variance of a Poisson random variable with mean $np_x$. $\qquad\square$

## C.3 Good-Turing estimators

To bound the regret corresponding to the Good-Turing estimator, we need few auxiliary results. The next set of equations follow from results in [13], For any $n$ and $t$,

$$\mathbb{E}[S_t] = \frac{t+1}{n} \cdot \mathbb{E}[\Phi_{t+1}]. \tag{8}$$

$$\mathrm{Var}(S_t) \leq \frac{(t+1)(t+2)}{n^2} \cdot \mathbb{E}[\Phi_{t+2}]. \tag{9}$$

$$\mathbb{E}\left[\left(S_t - \frac{(t+1)\Phi_{t+1}}{n}\right)^2\right] \leq \frac{(t+1)(t+2)\mathbb{E}[\Phi_{t+2}]}{n^2} + \frac{(t+1)^2\mathbb{E}[\Phi_{t+1}]}{n^2}. \tag{10}$$

The next lemma relates $\mathbb{E}[\Phi_{t+1}]$ to $\mathbb{E}[\Phi_t]$.

**Lemma 14.** *For any $n$ and $t \geq 1$,*

$$\mathbb{E}[\Phi_{t+1}] \leq \mathbb{E}[\Phi_t]\left(\frac{2}{t}\log n + \frac{t}{t+1}\right) + \frac{1}{t+1}.$$

*Proof.* Let $r \geq \frac{t}{t+1}$.

$$\mathbb{E}[\Phi_{t+1}] = \mathbb{E}\left[\sum_x 1_{N_x=t+1}\right]$$

$$= \sum_x e^{-np_x} \frac{(np_x)^{t+1}}{(t+1)!}$$

$$= \sum_x \frac{n}{t+1} \cdot e^{-np_x} \frac{(np_x)^t}{t!} p_x$$

$$= \sum_{x:np_x \leq r(t+1)} \frac{n}{t+1} \cdot e^{-np_x} \frac{(np_x)^t}{t!} p_x + \sum_{x:np_x > r(t+1)} \frac{n}{t+1} \cdot e^{-np_x} \frac{(np_x)^t}{t!} p_x$$

$$\overset{(a)}{\leq} r \sum_{x:np_x \leq r(t+1)} e^{-np_x} \frac{(np_x)^t}{t!} + \sum_{x:np_x > r(t+1)} \frac{n}{t+1} e^{-r(t+1)} \frac{(r(t+1))^t}{t!} p_x$$

$$\leq r \sum_x e^{-np_x} \frac{(np_x)^t}{t!} + \sum_x \frac{n}{t+1} e^{-r(t+1)} \frac{(r(t+1))^t}{t!} p_x$$

$$\overset{(b)}{\leq} r \sum_x e^{-np_x} \frac{(np_x)^t}{t!} + \sum_x \frac{n}{t+1} e^{-rt/2} p_x$$

$$\leq r\mathbb{E}[\Phi_t] + \frac{n}{t+1} e^{-\frac{rt}{2}}.$$

$(a)$ follows from the fact that second term is a decreasing as a function of $np_x$ in the range $[r(t+1), \infty)$. $(b)$ follows from the fact that

$$e^{-r(t+1)} \frac{(r(t+1))^t}{t!} = e^{-rt} r^t \cdot e^{-t} \frac{(t+1)^t}{t!} \leq e^{-rt} r^t \leq e^{-rt/2}.$$

Choosing $r = \frac{2}{t} \log n + \frac{t}{t+1}$, yields

$$\mathbb{E}[\Phi_{t+1}] \leq \mathbb{E}[\Phi_t] \left(\frac{2}{t} \log n + \frac{t}{t+1}\right) + \frac{1}{t+1}.$$

$\square$

The final auxiliary lemma bounds the inverse moment of Poisson binomial distributions.

**Lemma 15.** *Let $X_i$ for $1 \leq i \leq n$ be Bernoulli random variables, then*

$$\mathbb{E}\left[\frac{1}{\sum_{i=1}^n X_i + 1}\right] \leq \frac{1}{\sum_{i=1}^n \mathbb{E}[X_i]}.$$

*Proof.* Let $s_i = \mathbb{E}[X_i]$. We show that of all tuples $s_1, s_2, \ldots, s_n$ such that $\sum_{i=1}^n s_i = ns$, the one that maximizes the expectation is $s_i = s, \forall i$. Suppose for some $i, j$, $s_i > s_j$, we show that if we decrease $s_i$ and increase $s_j$ keeping the sum same, then the expectation increases. Let $Y = 1 + \sum_{k \notin \{i,j\}} X_k$. For any instance of $X^n$, taking expectation with respect to only $X_i$ and $X_j$.

$$\mathbb{E}\left[\frac{1}{X_i + X_j + Y} \mid Y\right] = \frac{(1-s_i)(1-s_j)}{Y} + \frac{s_i(1-s_j) + (1-s_i)s_j}{Y+1} + \frac{s_i s_j}{Y+2}$$

$$= \frac{1}{Y} + (s_i + s_j)\left(\frac{1}{Y+1} - \frac{1}{Y}\right) + s_i s_j \frac{2}{Y(Y+1)(Y+2)}.$$

Thus if we decrease $s_i$ and increase $s_j$ (keeping $s_i + s_j$ fixed), then $s_i s_j$ increases and hence the expectation increases. Hence the maximum occurs when $s_i = s_j$ for all $i, j$ and

$$\mathbb{E}\left[\frac{1}{\sum_{i=1}^n X_i + 1}\right] \leq \mathbb{E}\left[\frac{1}{Z+1}\right],$$

where $Z$ is a binomial random variable with parameters $n$ and $s = \sum_{i=1}^{n} \mathbb{E}[X_i]/n$.

The expectation can be bounded as

$$\mathbb{E}\left[\frac{1}{Z+1}\right] = \sum_{j=0}^{n} \frac{1}{j+1} \binom{n}{j} s^j (1-s)^{n-j}$$

$$= \frac{1}{(n+1)s} \sum_{j=0}^{n} \binom{n+1}{j+1} s^{j+1} (1-s)^{n+1-(j+1)}$$

$$\leq \frac{1}{(n+1)s}$$

$$\leq \frac{1}{ns}$$

$$= \frac{1}{\sum_{i=1}^{n} \mathbb{E}[X_i]}.$$

$\square$

Using the above lemma, we first bound the expectation of $S_t^2/(\Phi_{t+1}+1)$.

**Lemma 16.** *For any $n$ and $t$, if $\mathbb{E}[\Phi_{t+1}] > 2$, then*

$$\mathbb{E}\left[\frac{S_t^2}{\Phi_{t+1}+1}\right] \leq \frac{\mathbb{E}[S_t^2]}{\mathbb{E}[\Phi_{t+1}]-1}.$$

*Proof.* We first observe that for any $x$,

$$\mathbb{E}[1_{N_x=t+1}] = e^{-np_x} \frac{(np_x)^{t+1}}{(t+1)!} \leq e^{-t-1} \frac{(t+1)^{t+1}}{(t+1)!} \leq \frac{1}{e}. \tag{11}$$

Since $S_t = \sum_x p_x 1_{N_x=t}$ and $\Phi_{t+1} = \sum_x 1_{N_x=t+1}$,

$$\frac{S_t^2}{\Phi_{t+1}+1} = \frac{\sum_x \sum_y p_x p_y 1_{N_x=t} 1_{N_y=t}}{\sum_z 1_{N_z=t+1}+1} = \sum_x \sum_y \frac{p_x p_y 1_{N_x=t} 1_{N_y=t}}{\sum_{z:z\neq x, z\neq y} 1_{N_z=t+1}+1},$$

where the equality follows from the fact that symbol cannot appear both $t$ and $t+1$ times thus only one of $1_{N_x=t}$ and $1_{N_x=t+1}$ can be 1. The numerator and the denominator of the terms on RHS are independent of each other, hence

$$\mathbb{E}\left[\frac{p_x p_y 1_{N_x=t} 1_{N_y=t}}{\sum_z 1_{N_z=t+1}+1}\right] = \mathbb{E}\left[\frac{p_x p_y 1_{N_x=t} 1_{N_y=t}}{\sum_{z:z\neq x, z\neq y} 1_{N_z=t+1}+1}\right]$$

$$= \mathbb{E}\left[p_x p_y 1_{N_x=t} 1_{N_y=t}\right] \mathbb{E}\left[\frac{1}{\sum_{z:z\neq x, z\neq y} 1_{N_z=t+1}+1}\right]$$

$$\overset{(a)}{\leq} \frac{\mathbb{E}\left[p_x p_y 1_{N_x=t} 1_{N_y=t}\right]}{\sum_{z:z\neq x, z\neq y} \mathbb{E}[1_{N_z=t+1}]}$$

$$\overset{(b)}{\leq} \frac{\mathbb{E}\left[p_x p_y 1_{N_x=t} 1_{N_y=t}\right]}{\mathbb{E}[\Phi_{t+1}-1]},$$

$(a)$ follows from Lemma 15 and $(b)$ follows from Equation (11) as

$$\sum_{z:z\neq x, z\neq y} \mathbb{E}[1_{N_z=t+1}] = \sum_z \mathbb{E}[1_{N_z=t+1}] - \mathbb{E}[1_{N_x=t+1}] - \mathbb{E}[1_{N_y=t+1}] \geq \mathbb{E}[\Phi_{t+1}] - 1.$$

Summing over $x$ and $y$ results in the lemma. $\square$

We now have all the tools to bound the error of the Good-Turing estimator. We divide the set of values into two groups, depending on the value of $\mathbb{E}[\Phi_{t+1}]$.

**Lemma 17.** *For any $n$ and $t$ if $\mathbb{E}[\Phi_{t+1}] \leq 2$, then*
$$\mathbb{E}\left[\frac{(S_t - (t+1)(\Phi_{t+1}+1)/n)^2}{(\Phi_{t+1}+1)(t+1)/n}\right] \leq \frac{5t}{n} + \frac{4\log n}{n}\left(\frac{t+2}{t+1}\right) + \frac{6}{n}.$$

*Proof.* Let $Z = S_t - (t+1)\Phi_{t+1}/n$.

$$\mathbb{E}\left[\left(Z - \frac{t+1}{n}\right)^2\right] \overset{(a)}{=} \mathbb{E}[Z^2] + \frac{(t+1)^2}{n^2}$$

$$\overset{(b)}{\leq} \frac{(t+1)(t+2)\mathbb{E}[\Phi_{t+2}]}{n^2} + \frac{(t+1)^2\mathbb{E}[\Phi_{t+1}]}{n^2} + \frac{(t+1)^2}{n^2}$$

$$\overset{(c)}{\leq} 2\frac{(t+1)(t+2)}{n^2} \cdot \left(\frac{2\log n}{t+1} + \frac{t+1}{t+2}\right) + \frac{(t+1)(t+2)}{n^2(t+2)} + \frac{3(t+1)^2}{n^2}.$$

Equation (8) implies $Z$ is a zero mean random variable and hence $(a)$. Equation (10) implies $(b)$ and $(c)$ follows by Lemma 14 and the fact that $\mathbb{E}[\Phi_{t+1}] \leq 2$. Hence,

$$\mathbb{E}\left[\frac{(Z - (t+1)/n)^2}{(\Phi_{t+1}+1)(t+1)/n}\right] \leq \frac{\mathbb{E}[(Z-(t+1)/n)^2]}{(t+1)/n}$$

$$\leq \frac{2(t+2)}{n} \cdot \left(\frac{2\log n}{t+1} + \frac{t+1}{t+2}\right) + \frac{1}{n} + \frac{3(t+1)}{n}$$

$$= \frac{5t}{n} + \frac{4\log n(t+2)}{n(t+1)} + \frac{6}{n}.$$

$\square$

**Lemma 18.** *For any $n$ and $t$ if $\mathbb{E}[\Phi_{t+1}] > 2$, then*
$$\mathbb{E}\left[\frac{(S_t - (t+1)(\Phi_{t+1}+1)/n)^2}{(\Phi_{t+1}+1)(t+1)/n}\right] \leq \frac{5t}{n} + \frac{4\log n}{n}\left(\frac{t+2}{t+1}\right) + \frac{6}{n}.$$

*Proof.*
$$\frac{(S_t - (t+1)(\Phi_{t+1}+1)/n)^2}{(\Phi_{t+1}+1)(t+1)/n} = \frac{S_t^2}{(\Phi_{t+1}+1)(t+1)/n} + \frac{(t+1)(\Phi_{t+1}+1)}{n} - 2S_t.$$

Thus by Equation (8),
$$\mathbb{E}\left[\frac{(S_t - (t+1)(\Phi_{t+1}+1)/n)^2}{(\Phi_{t+1}+1)(t+1)/n}\right] = \mathbb{E}\left[\frac{S_t^2}{(\Phi_{t+1}+1)(t+1)/n}\right] - \frac{(t+1)\mathbb{E}[\Phi_{t+1}]}{n} + \frac{t+1}{n}. \quad (12)$$

By Lemma 16 and Equations (8), (9),
$$\mathbb{E}\left[\frac{S_t^2}{(\Phi_{t+1}+1)(t+1)/n}\right] \leq \frac{\mathbb{E}[S_t^2]}{\mathbb{E}[\Phi_{t+1}-1](t+1)/n}$$

$$\leq \frac{t+1}{n}\frac{\mathbb{E}[\Phi_{t+1}]^2}{\mathbb{E}[\Phi_{t+1}-1]} + \frac{t+2}{n}\frac{\mathbb{E}[\Phi_{t+2}]}{\mathbb{E}[\Phi_{t+1}-1]}.$$

Substituting the above equation in Equation (12) and simplifying,
$$\mathbb{E}\left[\frac{(S_t - (t+1)(\Phi_{t+1}+1)/n)^2}{(\Phi_{t+1}+1)(t+1)/n}\right] \leq \frac{(t+1)\mathbb{E}[\Phi_{t+1}] + (t+2)\mathbb{E}[\Phi_{t+2}]}{n\mathbb{E}[\Phi_{t+1}-1]} + \frac{t+1}{n}$$

$$\overset{(a)}{\leq} 2\frac{(t+1)\mathbb{E}[\Phi_{t+1}] + (t+2)\mathbb{E}[\Phi_{t+2}]}{n\mathbb{E}[\Phi_{t+1}]} + \frac{t+1}{n}$$

$$\overset{(b)}{\leq} 2\left(\frac{t+1}{n} + \frac{t+2}{n}\left(\frac{2\log n}{t+1} + \frac{t+1}{t+2} + \frac{1}{2(t+2)}\right)\right) + \frac{t+1}{n}$$

$$= \frac{5t}{n} + \frac{4\log n}{n}\left(\frac{t+2}{t+1}\right) + \frac{6}{n}.$$

Since $\mathbb{E}[\Phi_{t+1}] \geq 2$, $\mathbb{E}[\Phi_{t+1}] - 1 \geq \mathbb{E}[\Phi_{t+1}]/2$ and hence $(a)$. Lemma 14 implies $(b)$. $\square$

Combining the above two lemmas results in

**Lemma 19.** *For any $t_0 \geq 1$,*

$$\sum_{t=0}^{t_0-1} \mathbb{E}\left[\frac{(S_t - (t+1)(\Phi_{t+1}+1)/n)^2}{(\Phi_{t+1}+1)(t+1)/n}\right] \leq \frac{5t_0^2}{2n} + \frac{4\log n}{n}(t_0 + \log t_0 + 1) + \frac{7t_0}{2n}.$$

*Proof.* By Lemmas 17 and 18, regardless of the value of $\mathbb{E}[\Phi_{t+1}]$,

$$\mathbb{E}\left[\frac{(S_t - (t+1)(\Phi_{t+1}+1)/n)^2}{(\Phi_{t+1}+1)(t+1)/n}\right] \leq \frac{5t}{n} + \frac{4\log n}{n}\left(\frac{t+2}{t+1}\right) + \frac{6}{n}.$$

Summing the above expression for $0 \leq t \leq t_0 - 1$ results in the lemma. $\square$

Substituting the results from Lemmas 13 and 19 in Lemma 12,

$$\mathbb{E}[D(S\|\hat{S})] \leq \frac{1}{t_0} + \frac{5t_0^2}{2n} + \frac{4\log n}{n}(t_0 + \log t_0 + 1) + \frac{7t_0}{2n}.$$

Substituting $t_0 = n^{1/3}/5^{1/3}$ results in Theorem 1.

$$r_{\text{poi}(n)}^{\text{nat}}(q', \Delta_k) \leq \max_{p \in \Delta_k} \mathbb{E}[D(S\|\hat{S})] \leq \frac{2.6}{n^{1/3}} + \frac{2.4\log n(n^{1/3} + \log n + 1)}{n} + \frac{2.1}{n^{2/3}} \leq \frac{3 + o_n(1)}{n^{1/3}}.$$

## D  Proof of Theorem 3

To lower bound $r_n^{\mathbb{P}_\sigma}(\Delta_k)$ it is sufficient to lower bound $r_n^{\mathbb{P}_\sigma}(\mathcal{P})$ for any subset $\mathcal{P} \subseteq \Delta_k$. We construct a subset $\mathcal{P}$ by considering a set of distributions $\{p^{\bar{v}} : \bar{v} \in \{-1,1\}^{m-1}\}$ and all their possible permutations. The lower bound argument uses Fano's inequality and Gilbert Varshamov bounds.

We choose $\mathcal{P}$ to be the set of distributions whose probability multiset are close to that of a distribution $p^0$, where $p^0$ is defined as follows.

Let $c$ be a sufficiently large constant. Let $m$ be the largest odd number less than $\min(k, (n/(c^2 \log^2 n))^{1/3})$. Let $p^0$ be the following distribution. For $1 \leq i \leq m-1$,

$$p_i^0 = \frac{\log n}{6n}\sqrt{\frac{c^2 n}{m}}\left(\sqrt{\frac{n}{c^2 m \log^2 n}} + i\right)$$

and $p_m^0 = 1 - \sum_{i=1}^{m-1} p_i^0$. Observe that for all $1 \leq i \leq m-1$, $1/(6m) \leq p_i^0 \leq 1/(3m)$ and $p_m^0 \geq 2/3$.

We choose the close-by distributions as follows. Let $\epsilon = \sqrt{\frac{c^*}{mn}}$, where $c^*$ is some sufficiently small constant. For a binary vector $\bar{v} \in \{-1,1\}^{m-1}$, let $p^{\bar{v}}$ be the distribution such that $p_i^{\bar{v}} = p_i^0 + \bar{v}_i\epsilon$ for $1 \leq i \leq m-1$ and $p^{\bar{v}}(m) = 1 - \sum_{i=1}^{m-1} p_i^{\bar{v}}$. Note that by the properties of $p^0$ and $\epsilon$, $p^{\bar{v}}$ is a valid distribution for every $\bar{v}$. Let $\mathcal{C}$ be the largest subset of $\{-1,1\}^{m-1}$ such that for every $\bar{v} \in \mathcal{C}$, $\sum_i \bar{v}_i = 0$ and for every pair $\bar{v}, \bar{v}' \in \mathcal{C}$, $\sum_i |\bar{v}_i - \bar{v}_i'| \geq c'(m-1)$ for some constant $c'$. The following variation of Gilbert Varshamov lemma lower bounds size of $\mathcal{C}$.

**Lemma 20.** *There exists a set of vectors $\mathcal{C}$ over $\{-1,1\}^{m-1}$ of size $2^{c'' \cdot (m-1)}$ such that the minimum hamming distance between any two vectors is $\geq c'(m-1)$ for some universal constants $c' > 0, c'' > 0$ and $\sum_i \bar{v}_i = 0$ for all $\bar{v} \in \mathcal{C}$.*

Let $\mathcal{P}' = \{p^{\bar{v}} : \bar{v} \in \mathcal{C}\}$ and $P_{\bar{v}} = \{p^{\bar{v}}(\sigma(\cdot)) : \sigma \in \Sigma^{m-1}\}$ be the set of all permutations of a distribution $p^{\bar{v}}$, i.e., all distributions with the same multiset as $p^{\bar{v}}$. Let

$$\mathcal{P} = \cup_{\bar{v} \in \mathcal{C}} P_{\bar{v}}.$$

We first bound the regret of the induced permutation class $P_{\bar{v}}$ that contains all permutations of a distribution $p^{\bar{v}}$.

**Lemma 21.** *For every induced permutation class $P_{\bar{v}}$,*

$$r_n(P_{\bar{v}}) \leq \frac{1}{n}.$$

*Proof.* We prove the bound by constructing an estimator $q$. Consider the estimator $q$ which sorts the multiplicities and assigns the $i^{th}$-frequently occurred symbol probability $p_i^{\bar{v}}$. Since this is a natural estimator, it occurs the same loss for all distributions in $P_{\bar{v}}$ and hence,

$$
\begin{aligned}
r_n(P_{\bar{v}}) &\leq \max_{p \in P_{\bar{v}}} \mathbb{E}[D(p\|q)] \\
&= \mathbb{E}[D(p^{\bar{v}}\|q)] \\
&\overset{(a)}{\leq} \Pr(\exists i,j : N_i > N_j,\, p_i^{\bar{v}} < p_j^{\bar{v}}) \log n \\
&\overset{(b)}{\leq} \binom{m}{2} e^{-2\log n} \log n \\
&\leq \frac{1}{n}.
\end{aligned}
$$

$(a)$ follows from the fact that the estimator makes an error only if two multiplicities cross over and if it does make an error, the maximum KL divergence is at most $\log(p_{\max}/p_{\min}) \leq \log n$. Since probabilities for any two symbols $i$ and $j$ differ by at least $\frac{\log n}{6n} \cdot \sqrt{\frac{c^2 n}{m}}$ and the probabilities themselves lie between $1/(6m)$ and $1/(3m)$, by choosing a sufficiently large $c$, the cross over probability can be bounded by $e^{-2\log n}$ using the Chernoff bound and hence $(b)$. $\qquad\square$

We now lower bound the KL divergence between $p^{\bar{v}}$ and $p^{\bar{v}'}$ for every pair of vectors $\bar{v}$ and $\bar{v}'$. Let the Hamming distance between two vectors $\bar{v}$ and $\bar{v}'$ be $\|\bar{v} - \bar{v}'\|_1 = \sum_{i=1}^{m-1} |\bar{v}_i - \bar{v}_i'|$.

**Lemma 22.** *For two distributions $p^{\bar{v}}$ and $p^{\bar{v}'}$ in $\mathcal{P}'$,*

$$\frac{1}{8}\left(c'\sqrt{\frac{mc^*}{n}}\right)^2 \leq \frac{1}{2}\|p^{\bar{v}} - p^{\bar{v}'}\|_1^2 \leq D(p^{\bar{v}}\|p^{\bar{v}'}) \leq \frac{48mc^*}{n}.$$

*Proof.*

$$
\begin{aligned}
D(p^{\bar{v}}\|p^{\bar{v}'}) &\overset{(a)}{\leq} \sum_{i=1}^{m} \frac{(p_i^{\bar{v}} - p_i^{\bar{v}'})^2}{p_i^{\bar{v}'}} \\
&\overset{(b)}{\leq} 2\sum_{i=1}^{m} \frac{(p_i^{\bar{v}} - p_i^{\bar{v}'})^2}{p_i^0} \\
&\leq 2\sum_{i=1}^{m-1} \frac{(\bar{v}_i - \bar{v}_i')^2(\sqrt{c^*/nm})^2}{1/(6m)} \\
&\leq 12\sum_{i=1}^{m-1} \frac{(\bar{v}_i - \bar{v}_i')^2 c^*}{n} \\
&\leq 24\sum_{i=1}^{m-1} \frac{|\bar{v}_i - \bar{v}_i'|c^*}{n} \\
&= \frac{24\|\bar{v} - \bar{v}'\|_1 c^*}{n} \\
&\leq \frac{48mc^*}{n}.
\end{aligned}
$$

$(a)$ follows from bounding the KL divergence by the Chi-squared distance and $(b)$ follows from the fact that $\epsilon \ll 1/m$. For the lower bound,

$$
\begin{aligned}
D(p^{\bar{v}}||p^{\bar{v}'}) \overset{(a)}{\geq} & \frac{1}{2}||p^{\bar{v}} - p^{\bar{v}'}||_1^2 \\
= & \frac{1}{2}\left(\frac{||\bar{v} - \bar{v}'||_1\sqrt{c^*}}{\sqrt{mn}}\right)^2 \\
\overset{(b)}{\geq} & \frac{1}{2}\left(\frac{c'(m-1)\sqrt{c^*}}{\sqrt{mn}}\right)^2 \\
\overset{(c)}{\geq} & \frac{1}{8}\left(c'\sqrt{\frac{mc^*}{n}}\right)^2,
\end{aligned}
$$

where $(a)$ follows from Pinsker's inequality, $(b)$ follows by construction, and $m - 1 \geq 2$ and hence $(c)$. $\qquad\square$

We now state Fano's inequality for distribution estimation.

**Lemma 23.** *Let $p^1, p^2, \dots p^{r+1}$ be distributions such that $D(p^i||p^j) \leq \beta$ and $||p^i - p^j||_1 \geq \alpha$, for all $i, j$. For any estimator $q$,*

$$
\sup_i \mathbb{E}_i[||p^i - q||_1] \geq \frac{\alpha}{2}\left(1 - \frac{n\beta + \log 2}{\log r}\right).
$$

We now have all the tools for the lower bound.

*Proof of Theorem 3.* For every permutation subclass $P_{\bar{v}}$ in $\mathcal{P}$, by Lemma 21

$$
r_n(P_{\bar{v}}) \leq \frac{1}{n}.
$$

Thus,

$$
\begin{aligned}
r_n^{\mathbb{P}_\sigma}(\mathcal{P}) = & \min_q \max_{\bar{v}}\left(\max_{p \in P_{\bar{v}}} r_n(q, p) - r_n(P_{\bar{v}})\right) \\
\geq & \min_q \max_{\bar{v}}\left(\max_{p \in P_{\bar{v}}} r_n(q, p) - \frac{1}{n}\right) \\
= & \min_q \max_{p \in \mathcal{P}} r_n(q, p) - \frac{1}{n} \\
= & \min_q \max_{p \in \mathcal{P}} \mathbb{E}[D(p||q)] - \frac{1}{n} \\
\overset{(a)}{\geq} & \min_q \max_{p \in \mathcal{P}'} \mathbb{E}[D(p||q)] - \frac{1}{n} \\
\overset{(b)}{\geq} & \min_q \max_{p \in \mathcal{P}'} \mathbb{E}\left[\frac{||p - q||_1^2}{2}\right] - \frac{1}{n} \\
\overset{(c)}{\geq} & \min_q \max_{p \in \mathcal{P}'} \frac{1}{2}\mathbb{E}\left[||p - q||_1\right]^2 - \frac{1}{n} \\
\overset{(d)}{\geq} & \Omega\left(\frac{m}{n}\right) - \frac{1}{n} \\
\geq & \Omega\left(\frac{m}{n}\right).
\end{aligned}
$$

$\mathcal{P}' \subset \mathcal{P}$, hence $(a)$. $(b)$ follows from Pinsker's inequality and $(c)$ follows from convexity. By construction, for every pair of distributions in $\mathcal{P}'$, $\beta = D(p||p') \leq 48c^*m/n$ and $\alpha = ||p - p'||_1 \geq \Omega(\sqrt{m/n})$ (Lemma 22). Furthermore by Lemma 20, $\mathcal{P}'$ has $r+1 = 2^{c''(m-1)}$ distributions. Setting $c^*$ to be a sufficiently small constant and applying Lemma 23 to $\mathcal{P}'$ with the above values of $\alpha, \beta$, and $r$ results in $(d)$. Substituting the value of $m$ in the above equation results in the Theorem. $\qquad\square$