[Reviews · NeurIPS 2015]

Submitted by Assigned_Reviewer_1

The paper gives justification for the widespread use of the Good-Turing estimator for discrete distribution estimation through minimax regret analysis with two comparator classes. The paper obtains competitive regret bounds that lead to a more accurate characterization of the performance of the the Good-Turing estimators and in some cases is much better than the best known risk bounds. The comparator classes considered are estimators with knowledge of the distribution up to permutation, and estimators with full knowledge of the distribution, but with the constraint that the must assign the same probability mass to symbols appearing with the same frequencies.

While I like the exposition in the paper, I think there are several nice results in the appendix that should be highlighted that also clear up some confusions in the paper. For example, the fact that the optimal natural estimator can be explicitly computed and its risk can be bounded is very useful. First, it means that we actually obtain an "instance-specific" risk bound for the Good-Turing estimator by combining Lemma 10 with Theorem 2. It would be nice to give examples of this bound being much tighter than the best log(k/n) bound on the minimax risk.

Major comments: 1. In Figure 1, what is the estimator designed with prior knowledge of the partition? Is the red line a theoretical curve or an empirical one? Is there any analysis for the case where the partition is known? Roughly, my question is whether that red curve actually lower bounds the minimax risk here. I know you are using the lower bound r_n(\Delta_k) \ge r_n(P), but I am not sure how you are calculating the min_q in the definition of r_n(P). It is not sufficient to use an upper bound (i.e. a sub-optimal estimator) if the claim is that the red curve lower bounds the r_n(\Delta_k).

I think this is answered in the proof of Lemma 4 in the appendix but it should be made clear in the main paper.

2. In the experiments it would be helpful to make it clear what things are known to the estimators and what aren't. I believe the quantity S(t) is not known to fully data-driven estimators and only to the comparator in the natural class. Is that correct? 3. More generally, I noticed that some notation is defined only in the appendix and not in the main paper and vice versa. It would be nicer and aid readability if all of the notation was in the main paper (and maybe even remind the reader from time to time).

Minor comments: 1. In the abstract you use k and n but those variables have not been defined yet.

2. "The distributions resulting in r_n(\Delta_k)." I think you mean to say the distributions achieving the max in the definition. 3. The claim that "which in turn is better than the performance of any data-driven estimator designed without prior information" is technically not true for the second comparator since in theory there could be an estimator which does not assign the same estimate to symbols with the same frequency. I do believe the prior information makes the problem easier but technically there could be a better estimator that does not obey this constraint. 4. Some typos in the first paragraph of section 2.3. Sentence beginning with "for example..." is not technically a sentence. No space after period. 5. Typo line 227. 6. n(x) undefined in the main text. I think it is the number of times symbol x appeared in the sample.
Summary: This paper makes a compelling case for the popularity of the Good-Turing estimator for discrete distribution estimation. I think the problem is important, the results seem non-trivial and significant (although I am not an expert in this area), and the presentation is coherent, although there are some ambiguities.

Submitted by Assigned_Reviewer_2

The authors present several notions of competitive optimality for estimating a pmf over a set of large cardinal. They show that a variant of the Good-Turing estimator is competitively optimal for any distribution.

The paper is well written, provides novel results which are illustrated by simulations.

Summary: Very good paper overall.

Submitted by Assigned_Reviewer_3

This paper considers the estimation of distributions over large alphabets. The authors formulate the regret relative to the oracle with prior knowledge of the data-generating distribution up to a permutation and the regret relative to the best natural estimator with prior knowledge of the exact distribution. Then they prove that variants of Good-Turing estimators are nearly optimal in the two relative performance measures.

This is a solid theoretical contribution to the universal estimation of distributions over large alphabets.

p.4, l.204: The reference [24] is cited as a work dealing with l_1 distance. I wonder if the orders of the regrets of authors' and reference's are comparable since different distance measures are used.

p.5, l.268: It is mentioned that the regret approaches the upper bound for some probability multisets. Does the theory imply anything on what are probability multisets like in such a case?

p.7, Fig.2: Some bounds have the form O(min(1/\sqrt{n}, k/n)). Is it possible to examine from the simulation results in which order of n the regret decreases?
Summary: This is a solid theoretical contribution to the universal estimation of distributions over large alphabets. I have a few minor concerns on which additional discussions might be nice.

Submitted by Assigned_Reviewer_4

As the authors discuss, previous work [4] showed that add-constant estimators are min-max near-optimal (i.e. result

of equation (3).

The authors say that in practice add-constant estimators are not very good since they essentially tend to prefer near-uniform estimates. This is reasonable and shows the limitations of the min-max criterion, since two estimators are only compared for the worst-case (which seems to be near-uniform distributions).

Perhaps a good analogy here is in comparing the speed of different vehicles: in a min-max sense (over all possible terrains) the tank is the fastest vehicle, since it will perform better than everything else in deep mud. This shows the limitation of the min-max criterion and how it can lead to incorrect conclusions.

(Edit after reading reply: This reviewer will be glad if the authors include the tank analogy in their paper. Attributing to 'Anonymous reviewer' would be sufficient. Tank manufacturers will rejoice since their vehicles are provably the fastest. )

The authors argue that none of the previous results explain why Good-Turing estimators perform well in real problems and propose two new criteria: a `local' min-max criterion and a 'comparing against natural estimators' criterion. The authors establish nice uniform optimality results for both criteria that justify the use of Good-Turing estimators. The obtained results are quite surprising and cool.

A result like (3) i.e. a 'tank-like' min-max optimality for Good-Turing is the starting point of this paper.

It was not clear if the authors believe that

1. This result is not true for Good-Turing and for that reason we need new criteria for comparing estimators.

2. It must be true but we can't prove it

Perhaps the authors could discuss this a bit more.

227: using experiments - (?) 234: 'poi(n)' should be explained that you mean poisson distributed and with the correct parameter
Summary: The paper discusses the problem of distribution estimation in the large-alphabet regime.

The authors establish novel results for Good-Turing estimators that show how GT are near-optimal in a novel competitive sense. These results were quite surprising.

The authors have also a nice background discussion on min-max optimality for different estimators and explain how it is perhaps the wrong evaluation metric for practical probability estimation.

The paper is very well-written and contains interesting new ideas.

Author Feedback
Author rebuttal: We thank all the reviewers for their positive and helpful comments. We incorporated the comments regarding notation, typos, and intuition in our draft and will submit the new version if the paper gets accepted. Here we address their major comments.

Reviewer 1:

1. First, it means that we actually obtain an "instance­specific" risk bound for the Good­Turing estimator by combining Lemma 10 with Theorem 2. It would be nice to give examples of this bound being much tighter than the best log(k/n) bound on the minimax risk.

We will add some trivial and non-trivial examples where the instance specific bound is much smaller than log (k/n). The simplest trivial example are singleton distributions of the form $(0,1,0,..)$. For these distributions the instance specific risk is 0 as after observing one symbol we know the distribution. This can be easily generalized to distributions with m possible symbols. There are also non-trivial examples, like power-law, that are also important in practice. We will elaborate on them in the version we upload.

2. In Figure 1, what is the estimator designed with prior knowledge of the partition? Is the red line a theoretical curve or an empirical one? Is there any analysis for the case where the partition is known? Roughly, my question is whether that red curve actually lower bounds the minimax risk here. I know you are using the lower bound r_n(\Delta_k) \ge r_n(P), but I am not sure how you are calculating the min_q in the definition of r_n(P). It is not sufficient to use an upper bound (i.e. a sub-optimal estimator) if the claim is that the red curve lower bounds the r_n(\Delta_k)...

This is a very helpful comment. Figure 1 is a qualitative illustration of our results. As the reviewer suggests, the red curve represents the lower bound, the optimal estimator designed with prior knowledge of the underlying mulitset, hence derived without using a sub-optimal estimator. This is indeed addressed in the proof of Lemma 4 where the estimator is called q'' and is shown to satisfy r_n(P) = max_{p \in P} D(p||q''). We will highlight q'' in the main paper.

3. In the experiments it would be helpful to make it clear what things are known to the estimators and what aren't. I believe the quantity S(t) is not known to fully data-driven estimators and only to the comparator in the natural class. Is that correct?

Yes, that is correct. All the estimators (Laplace, our Good-Turing, etc.) observe just the samples and are not aware of S(t) or any other distribution properties. Only the idealized best-natural estimator (red curve lower bound), is assumed to know S(t).

4. More generally, I noticed that some notation is defined only in the appendix and not in the main paper and vice versa. It would be nicer and aid readability if all of the notation was in the main paper (and maybe even remind the reader from time to time).

Thank you for pointing this out, we would incorporate all these changes.

Reviewer 2:

1. p.4, l.204: The reference [24] is cited as a work dealing with l_1 distance. I wonder if the orders of the regrets of authors' and reference's are comparable since different distance measures are used.

The regret of the reference is O(1/poly(log n)), ours is significantly lower, O(1/poly(n)).

2. p.5, l.268: It is mentioned that the regret approaches the upper bound for some probability multisets. Does the theory imply anything on what are probability multisets like in such a case?

This is a good question. Our results do not indicate which probability multisets approach the upper bound and which do not. Intuitively, one would imagine that more ``structured'' distributions will be further below the upper bound, but we do not yet have a way of formalizing or proving that.

3. p.7, Fig.2: Some bounds have the form O(min(1/\sqrt{n}, k/n)). Is it possible to examine from the simulation results in which order of n the regret decreases?

We had not considered this question. It would indeed be interesting to see which of the two parts of the upper bound is binding for moderate k and n.

Reviewer 3:

We thank the reviewer for pointing the nice analogy with vehicles. May we use it (happily with attribution..) in the future? :-)

1. A result like (3) i.e. a 'tank-like' min-max optimality for Good-Turing is the starting point of this paper...

This is a very good observation and we will mention it in the paper's final version. Theorem 1 indeed implies that the Good-Turing estimator achieves a result like (3). However the implied result would be up to a constant factor. We have not yet explored the exact constant in the Good-Turing estimation.

Reviewer 4:

We thank reviewer 4 for the comments and as stated before, we would move some material from supplementary to the main paper.

Reviewer 6:

We thank reviewer 6 for the positive and encouraging comments.